# ControlVideo: Conditional Control for Text-driven Video Editing and Beyond

## Abstract

This paper presents *ControlVideo* for text-driven video editing – generating a video that aligns with a given text while preserving the structure of the source video. Building on a pre-trained text-to-image diffusion model, ControlVideo enhances the fidelity and temporal consistency by incorporating additional conditions (such as edge maps), and fine-tuning the key-frame and temporal attention on the source video-text pair via an in-depth exploration of the design space. Extensive experimental results demonstrate that ControlVideo outperforms various competitive baselines by delivering videos that exhibit high fidelity w.r.t. the source content, and temporal consistency, all while aligning with the text. By incorporating Low-rank adaptation layers into the model before training, ControlVideo is further empowered to generate videos that align seamlessly with reference images. Moreover, ControlVideo can be readily extended to the more challenging task of long video editing, where maintaining long-range temporal consistency across hundreds of frames is crucial. To achieve this, we construct a fused ControlVideo by applying basic ControlVideo to overlapping short video segments and key frame videos and then merging them by defined weight functions. Empirical results corroborate its ability to create visually realistic videos spanning hundreds of frames.

## 1 Introduction

The endeavor of text-driven video editing is to generate videos derived from textual prompts and existing video footage, thereby reducing manual labor. This technology stands to significantly influence an array of fields such as advertising, marketing, and social media content. During this process, it is critical for the edited videos to *faithfully* preserve the content of the source video, maintain *temporal consistency* between generated frames, and *align* with the provided text. However, fulfilling all these requirements simultaneously poses substantial challenges. A further challenge arises when dealing with real-world videos that typically consist of hundreds of frames: how can *long-range temporal consistency* be maintained? Additionally, what if textual descriptions fail to convey the precise desired effects as intended by users, and users wish for the generated video to also *align* with reference images?

Previous research (Qi et al., 2023; Wang et al., 2023; Wu et al., 2022; Liu et al., 2023a) has made significant strides in text-driven video editing, capitalizing on advancements in large-scale text-to-image (T2I) diffusion models (Rombach et al., 2022; Ho et al., 2022; Ramesh et al., 2022) and image editing techniques (Hertz et al., 2023; Tumanyan et al., 2022; Parmar et al., 2023). However, despite these advancements, they still cannot address the aforementioned challenges: (1) empirical evidence (see Fig. 4) suggests that existing approaches still struggle with faithfully controlling the output while preserving temporal consistency, and (2) these approaches primarily focus on short video editing and do not explore how to maintain temporal consistency over extended durations.

To this end, we present *ControlVideo* for faithful and temporal consistent video editing, building upon a pre-trained T2I diffusion model. To enhance fidelity, we propose to incorporate visual conditions such as edge maps as additional inputs into T2I diffusion models to amplify the guidance from the source video. As ControlNet (Zhang & Agrawala, 2023) has been pre-trained alongside the diffusion model, we utilize it to process these visual conditions. Recognizing that various visual conditions encompass varying degrees of information from the source video, we engage in a comprehensive investigation of the suitability of different visual conditions for different scenes. This exploration

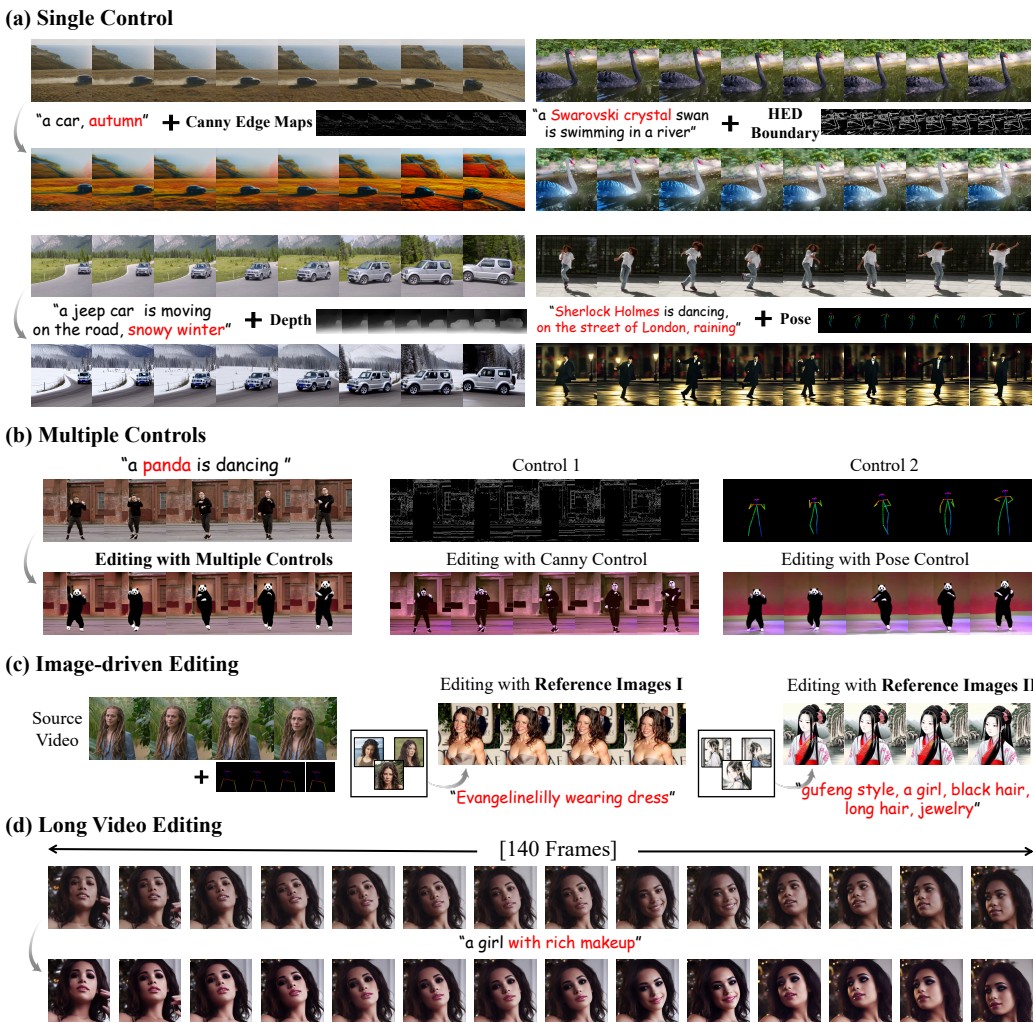

Figure 1: Main results of ControlVideo with (a) single control, (b) multiple controls, (c) image-driven video editing, and (d) long video editing.

naturally leads us to combine multiple controls to leverage their respective advantages. Furthermore, we transform the original spatial self-attention into key-frame attention, aligning all frames with a selected one, and incorporate temporal attention modules as extra branches in the diffusion model to improve faithfulness and temporal consistency further, which is designed by a systematic empirical study. Additionally, ControlVideo can generate videos that align with reference images by introducing Low-rank adaptation (LoRA) (Hu et al., 2021) layers on the diffusion model before training.

Empirically, we validate our method on 50 video-text pair data collected from the Davis dataset following previous works (Qi et al., 2023; Liu et al., 2023a; Wu et al., 2022) and the internet. We compare with frame-wise Stable Diffusion and SOTA text-driven video editing methods (Qi et al., 2023; Liu et al., 2023a; Wu et al., 2022) under objective metrics and a user study. In particular, following (Qi et al., 2023; Liu et al., 2023a) we use CLIP (Radford et al., 2021) to measure text-alignment and temporal consistency and employ SSIM to assess faithfulness. Extensive results demonstrate that ControlVideo outperforms various competitors by fulfilling three requirements of text-driven video editing simultaneously. Notably, ControlVideo can produce videos with extremely realistic visual quality and very faithfully preserve original source content while following the text guidance. For instance, ControlVideo can successfully make up a woman with maintaining her identity while all existing methods fail (see Fig. 4).

Furthermore, ControlVideo is readily extendable for the more challenging application: video editing for long videos that encompass hundreds of frames (see Sec. 3.2). To achieve this, we construct

a fused ControlVideo by applying basic ControlVideo to overlapping short videos and key frame videos and then merging them by defined weight functions at each denoising step. Intuitively, fusion with overlapping short videos encourages the overlapping frames to merge features from neighboring short videos, thereby effectively mitigating inconsistency issues between adjacent video clips. On the other hand, key frame video, which incorporates the first frame of each video segment, provides global guidance from the whole video, and thus fusion with it can improve long-range temporal consistency. Empirical results affirm ControlVideo's ability to produce videos spanning hundreds of frames, exhibiting a high degree of visual realism.

## 2 BACKGROUND

**Diffusion Models for Image Generation and Editing.** Let $q(\boldsymbol{x}_0)$ be the data distribution on $\mathbb{R}^D$. Diffusion models (Song et al., 2020b; Bao et al., 2021; Ho et al., 2020) gradually perturb data $\boldsymbol{x}_0 \sim q(\boldsymbol{x}_0)$ by a forward diffusion process:

$$q(\boldsymbol{x}_{1:T}) = q(\boldsymbol{x}_0)\prod_{t=1}^{T} q(\boldsymbol{x}_t|\boldsymbol{x}_{t-1}), \quad q(\boldsymbol{x}_t|\boldsymbol{x}_{t-1}) = \mathcal{N}(\boldsymbol{x}_t; \sqrt{\alpha_t}\boldsymbol{x}_{t-1}, \beta_t \boldsymbol{I}), \tag{1}$$

where $\beta_t$ is the noise schedule, $\alpha_t = 1 - \beta_t$ and is designed to satisfy $\boldsymbol{x}_T \sim \mathcal{N}(\boldsymbol{0}, \boldsymbol{I})$. The forward process $\{\boldsymbol{x}_t\}_{t\in[0,T]}$ has the following transition distribution:

$$q_{t|0}(\boldsymbol{x}_t|\boldsymbol{x}_0) = \mathcal{N}(\boldsymbol{x}_t|\sqrt{\bar{\alpha}_t}\boldsymbol{x}_0, (1 - \bar{\alpha}_t)\boldsymbol{I}), \tag{2}$$

where $\bar{\alpha}_t = \prod_{s=1}^{t} \alpha_s$. The data can be generated starting from $\boldsymbol{x}_T \sim \mathcal{N}(\boldsymbol{0}, \boldsymbol{I})$ through the reverse diffusion process, where the reverse transition kernel $q(\boldsymbol{x}_{t-1}|\boldsymbol{x}_t)$ is learned by a Gaussian model: $p_\theta(\boldsymbol{x}_{t-1}|\boldsymbol{x}_t) = \mathcal{N}(\boldsymbol{x}_{t-1}; \boldsymbol{\mu}_\theta(\boldsymbol{x}_t), \sigma_t^2 \boldsymbol{I})$. Ho et al. (2020) shows learning the mean $\boldsymbol{\mu}_\theta(\boldsymbol{x}_t)$ can be derived to learn a noise prediction network $\epsilon_\theta(\boldsymbol{x}_t, t)$ via a mean-squared error loss:

$$\min_\theta \mathbb{E}_{t,\boldsymbol{x}_0,\boldsymbol{\epsilon}}||\boldsymbol{\epsilon} - \epsilon_\theta(\boldsymbol{x}_t, t)||^2, \tag{3}$$

where $\boldsymbol{x}_t \sim q_{t|0}(\boldsymbol{x}_t|\boldsymbol{x}_0), \boldsymbol{\epsilon} \sim \mathcal{N}(\boldsymbol{0}, \boldsymbol{I})$. Deterministic DDIM sampling (Song et al., 2020a) generate samples starting from $\boldsymbol{x}_T \sim \mathcal{N}(\boldsymbol{0}, \boldsymbol{I})$ via the following iteration rule:

$$\boldsymbol{x}_{t-1} = \sqrt{\alpha_{t-1}}\frac{\boldsymbol{x}_t - \sqrt{1 - \alpha_t}\epsilon_\theta(\boldsymbol{x}_t, t)}{\sqrt{\alpha_t}} + \sqrt{1 - \alpha_{t-1}}\epsilon_\theta(\boldsymbol{x}_t, t). \tag{4}$$

Different from unconditional generation, image editing needs to preserve the content from the source image $\boldsymbol{x}_0$. Considering the reversible property of ODE, DDIM inversion (Song et al., 2020a) is adopted to convert a real image $\boldsymbol{x}_0$ to related inversion noise $\boldsymbol{x}_M$ by reversing the above process for faithful image editing:

$$\boldsymbol{x}_t = \sqrt{\alpha_t}\frac{\boldsymbol{x}_{t-1} - \sqrt{1 - \alpha_{t-1}}\epsilon_\theta(\boldsymbol{x}_{t-1}, t-1)}{\sqrt{\alpha_{t-1}}} + \sqrt{1 - \alpha_t}\epsilon_\theta(\boldsymbol{x}_{t-1}, t-1). \tag{5}$$

**Latent Diffusion Models and ControlNet.** To reduce computational cost, latent diffusion models (LDM, a.k.a Stable Diffusion) (Rombach et al., 2022) use an encoder $\mathcal{E}$ to transform $\boldsymbol{x}_0$ into low-dimensional latent space $\boldsymbol{z}_0 = \mathcal{E}(\boldsymbol{x}_0)$, which can be reconstructed by a decoder $\boldsymbol{x}_0 \approx \mathcal{D}(\boldsymbol{z}_0)$, and then learns the noise prediction network $\epsilon_\theta(\boldsymbol{z}_t, p, t)$ in the latent space, where $p$ is the textual prompts. The backbone for $\epsilon_\theta(\boldsymbol{z}_t, p, t)$ is the UNet (termed *main UNet*) that stacks several basic blocks. To enable models to learn additional conditions $c$, ControlNet (Zhang & Agrawala, 2023) adds a trainable copy of the encoder and middle blocks of the main UNet (termed *ControlNet*) to incorporate task-specific conditions on the locked Stable Diffusion. The outputs of ControlNet are then followed by a zero-initialization convolutional layer, which is subsequently added to the features of the main U-Net at the corresponding layer. We describe and visualize the detailed architecture of the main UNet and ControlNet in Appendix B.

## 3 METHODS

In this section, we present *ControlVideo* for faithful and temporally consistent text-driven video editing via an in-depth exploration of the design space (see Sec. 3.1). As shown in Figure 2, this is

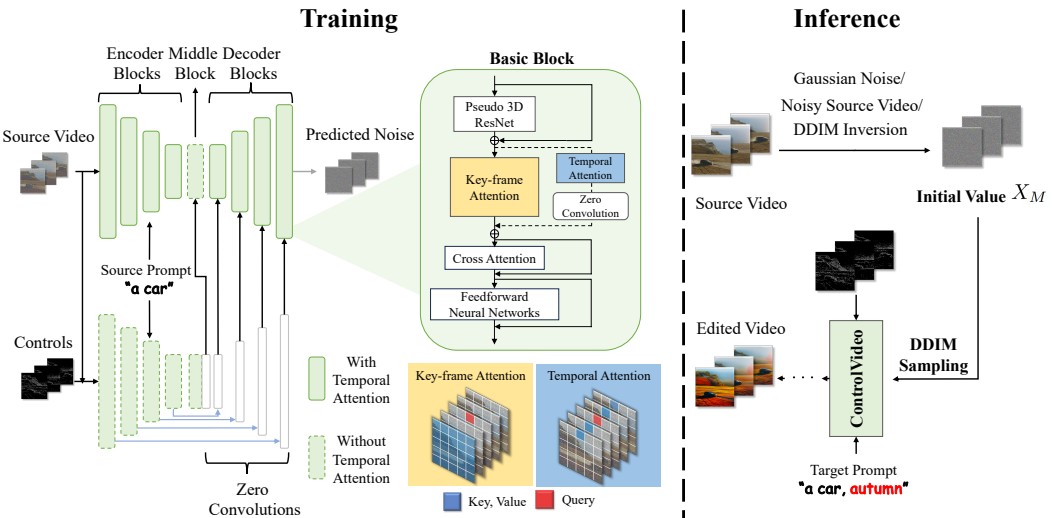

Figure 2: Flowchart of ControlVideo. ControlVideo builds on a pre-trained T2I diffusion model and enhances the fidelity and temporal consistency by incorporating additional conditions, fine-tuning the key-frame, and temporal attention on the source video-text pair. We can generate the edited video starting using DDIM sampling based on the target prompt starting from the initial value $X_M$.

achieved by incorporating additional conditions, fine-tuning the key-frame, and temporal attention on the source video-text pair. By incorporating Low-rank adaptation layers, ControlVideo is further empowered to generate videos that align seamlessly with reference images. Furthermore, in Sec. 3.2, we extend ControlVideo for the more challenging application: long video editing.

## 3.1 CONTROLVIDEO

### 3.1.1 ARCHITECTURE

In line with prior studies (Wu et al., 2022; Qi et al., 2023), we first replace the spatial kernel ($3 \times 3$) in 2D convolution layers with 3D kernel ($1 \times 3 \times 3$) to handle video inputs.

**Adding Visual Controls.** Recall that a key objective in text-driven video editing is to *faithfully* preserve the content of the source video. An intuitive approach is to generate edited videos starting from DDIM inversion $X_M$ in Eq. 5 to leverage information from $X_0$. However, despite the reversible nature of ODE, as depicted in Fig. 3, empirically, the combination of DDIM inversion and DDIM sampling significantly disrupts the structure of the source video. To enhance fidelity, we propose to introduce additional visual conditions $C = \{c^i\}_{i=1}^N$, such as edge maps for all frames, into the main UNet to amplify the source video's guidance: $\epsilon_\theta(X_t, C, p, t)$. Notably, as ControlNet(Zhang & Agrawala, 2023) has been pre-trained alongside the main UNet in Stable Diffusion, we utilize it to process these visual conditions $C$. Formally, let $h_u \in \mathbb{R}^{N \times d}$ and $h_c \in \mathbb{R}^{N \times d}$ denote the hidden features with dimension $d$ of the same layer in the main UNet and ControlNet, respectively. We combine these features by summation, yielding $h = h_u + \lambda h_c$, which is then fed into the decoder of the main UNet through a skip connection, with $\lambda$ serving as the control scale. As illustrated in Figure 3, the introduction of visual conditions to provide structural guidance from $X_0$ significantly enhances the faithfulness of the edited videos.

Further, given that different visual conditions encompass varying degrees of information derived from $X_0$, we comprehensively investigate *the advantages of employing different conditions*. As depicted in Figure 1, our findings indicate that conditions yielding detailed insights into $X_0$, such as edge maps, are particularly advantageous for attribute manipulation such as facial video editing, demanding precise control to preserve human identity. Conversely, conditions offering coarser insights into $X_0$, such as pose information, facilitate flexible adjustments to shape and background. This exploration naturally raises the question of whether we can combine *multiple controls* to leverage their respective advantages. To this end, we compute a weighted sum of hidden features derived from different controls, denoted as $h = h_u + \sum_i \lambda_i h_c$, and subsequently feed the fused features

into the decoder of the main UNet, where $\lambda_i$ represents the control scale associated with the $i$-th control. In situations where multiple controls may exhibit conflicts or inconsistencies, we can employ Grounding-DINO (Liu et al., 2023b) and SAM (Kirillov et al., 2023) or cross-attention map (Hertz et al., 2023) to generate a mask based on text and feed the masked controls into ControlVideo to enhance control synergy. As shown in Figure 1, Canny edge maps excel at preserving the background while having a limited impact on shape modification. In contrast, pose control facilitates flexible shape adjustments but may overlook other crucial details. By combining these controls, we can simultaneously preserve the background and effect shape modifications, demonstrating the feasibility of leveraging multiple controls in complex video editing scenarios.

**Key-frame Attention.** The self-attention in T2I diffusion models updates the features of each frame independently, resulting in temporal inconsistencies within the generated videos. To address this issue and improve *temporal consistency*, we propose to introduce a key frame that serves as a reference for propagating information throughout the video. Specifically, drawing inspiration from previous works (Wu et al., 2022), we transform the spatial self-attention in both main UNet and ControlNet into key-frame attention, aligning all frames with a selected reference frame. Formally, let $v^i \in \mathbb{R}^d$ represent the hidden features of the $i$-th frame, and let $k \in [1, N]$ denote the chosen key frame. The key-frame attention mechanism is defined as follows:

$$Q = W^Q v^i, K = W^K v^k, V = W^V v^k,$$

where $W^Q, W^K, W^V$ are the projected matrix. We initialize these matrices using the original self-attention weights to leverage the capabilities of T2I diffusion models fully. Empirically, we systematically study *the design of key frame, key and value selection in self-attention and fine-tuned parameters*. A detailed analysis is provided in Appendix C. In summary, we utilize the first frame as key frame, which serves as both the key and value in the attention mechanism, and we finetune the output projected matrix $W^O$ within the attention modules to enhance temporal consistency.

**Temporal Attention.** In pursuit of enhancing both the *faithfulness* and *temporal consistency* of the edited video, we introduce temporal attention modules as extra branches in the network, which capture relationships among corresponding spatial locations across all frames. Formally, let $v \in \mathbb{R}^{N \times d}$ denote the hidden features, the temporal attention is defined as follows:

$$Q = W^Q v, K = W^K v, V = W^V v.$$

Prior research (Singer et al., 2022) has benefited from extensive data to train temporal attention, a luxury we do not have in our one-shot setting. To address this challenge, we draw inspiration from the consistent manner in which different attention mechanisms model relationships between image features. Accordingly, we initialize temporal attention using the original spatial self-attention weights, harnessing the capabilities of the T2I diffusion model. After each temporal attention module, we incorporate a zero convolutional layer (Zhang & Agrawala, 2023) to retain the module's output prior before fine-tuning. Furthermore, we conduct a comprehensive study on *the incorporation of local and global positions for introducing temporal attention*. Detail analyses are provided in Appendix C. Concerning local positions, we find that the most effective placement is both before and within the self-attention mechanism within the transformer block. This choice is substantiated by the fact that the input in these two positions matches that of self-attention, serving as the initial weights for temporal attention. With self-attention location exhibits higher text alignment, ultimately making it our preferred choice. For global location, our main finding is that the effectiveness of positions is correlated with the amount of information they encapsulate. For instance, the main UNet responsible for image generation retains a full spectrum of information, outperforming the ControlNet, which focuses solely on extracting condition-related features while discarding others. As a result, we incorporate temporal attention alongside self-attention at all stages of the main UNet, with the exception of the middle block.

### 3.1.2 TRAINING AND SAMPLING FRAMEWORK

Let $C = \{c^i\}_{i=1}^N$ denote the visual conditions (e.g., Canny edge maps) for $X_0$ and $\epsilon_\theta(X_t, C, p, t)$ denote the ControlVideo network. Let $p_s$ and $p_t$ represent the source prompt and target prompt, respectively. Similar to Eq. 3, we finetune $\epsilon_\theta(X_t, C, p, t)$ on the source video-text pair $(X_0, p_s)$ using the mean-squared error loss, defined as follows:

$$\min_\theta \mathbb{E}_{t,\epsilon} ||\epsilon - \epsilon_\theta(X_t, C, p_s, t)||^2,$$

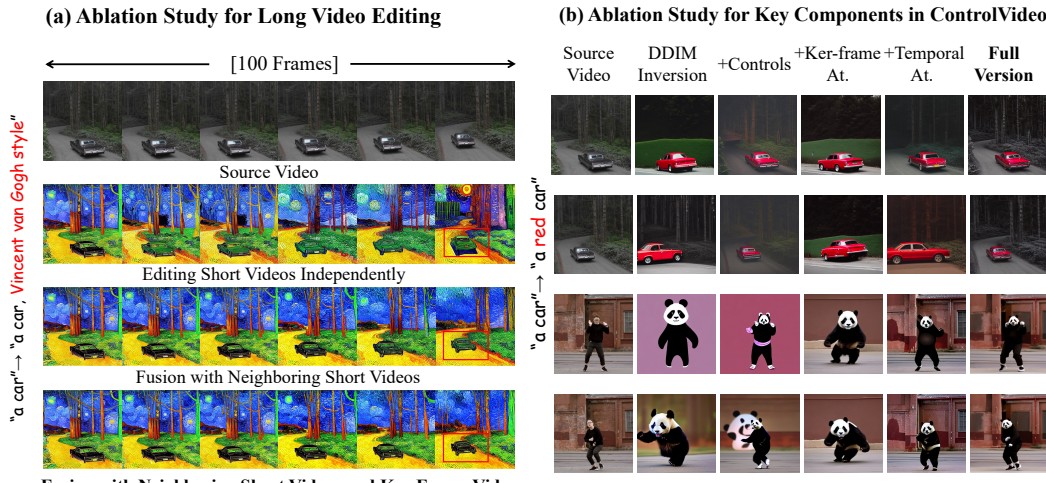

Figure 3: (a) Ablation studies for long video editing. See detailed analysis in Sec. 3.2. (b) Ablation studies for key components in ControlVideo. At. denote attention. See detailed analysis in Sec. 5.2.

where $\epsilon \sim \mathcal{N}(\mathbf{0}, \mathbf{I}), X_t \sim q_{t|0}(X_t|X_0)$. Note that during training, we exclusively optimize the parameters within the attention modules (as discussed in Sec. 3.1.1), while keeping all other parameters fixed.

**Choice of Initial Value** $X_M$**.** Built upon $\epsilon_\theta(X_t, C, p, t)$, we can generate the edited video starting from the initial value $X_M$ using DDIM sampling (Song et al., 2020a), based on the target prompt $p_t$. For $X_M$, we employ DDIM inversion as described in Eq. 5 for local editing tasks, such as attribute manipulation. For global editing such as style transfer, different from previous work (Wu et al., 2022; Qi et al., 2023), we can also start from noisy source video $X_M \sim q_{M|0}(X_M|X_0)$ using forward transition distribution in Eq. 2 with large $M$ and even $X_M \sim \mathcal{N}(\mathbf{0}, \mathbf{I})$ to improve editability because visual conditions have already provided structure guidance from $X_0$.

---

**Algorithm 1** Extended ControlVideo for Long Video Editing

---

**Require:** initial value $X_M$, controls $C$, short video length $L$, overlapped length $a$, fusion function $F(\cdot)$, weight $w$, model $\epsilon_\theta(\cdot, \cdot, \cdot, \cdot)$, prompt $p$
    $n = \lfloor N/(L-a) \rfloor + 1$                                  {number of short videos}
    **for** $t = M$ to $1$ **do**
        **for** $j = 1$ to $n$ **do**
            $\epsilon_\theta^j \leftarrow \epsilon_\theta(X_t^j, C^j, p, t)$                   {ControlVideo for each short video}
        **end for**
        $\hat{\epsilon}_\theta \leftarrow F(\epsilon_\theta^1, \ldots, \epsilon_\theta^n)$           {fusion with neighboring short videos via Eq. 7}
        $\epsilon_\theta^K \leftarrow \epsilon_\theta(X_t^K, C^K, p, t)$                  {ControlVideo for key frame video}
        $\epsilon_\theta \leftarrow wO(\epsilon_\theta^K) + (1-w)\hat{\epsilon}_\theta$          {fusion with key frame video via Eq. 8}
        $X_{t-1} \leftarrow \text{DDIM\_Sampling}(\epsilon_\theta, X_t, p, t)$         {denoising step in Eq. 4}
    **end for**
    **return** $X_0$

---

### 3.1.3 IMAGE-DRIVEN VIDEO EDITING

In certain scenarios, textual descriptions may fall short of fully conveying the precise desired effects from users. In such cases, users may wish for the generated video to also *align* with given reference images. Here, we show a simple way to extend ControlVideo for image-driven video editing. Specifically, we can first add the Low-rank adaptation (LoRA)(Hu et al., 2021) layer on the main UNet to facilitate the learning of concepts relevant to reference images and then freeze them to train ControlVideo following Sec. 3.1.2. Since the training for reference images and video is independent, we can flexibly utilize models in the community like CivitAI (Civ).

## 3.2 EXTENDED CONTROLVIDEO FOR LONG VIDEO EDITING

Although ControlVideo described in the above section has the appealing ability to generate highly temporal consistent videos, it is still difficult to deal with real-world videos that typically encompass hundreds of frames due to memory limitations. A straightforward approach to address this issue involves dividing the entire video into several shorter segments and applying ControlVideo to each segment with a strategy that initializes all frames with the same value $x_M^i = \epsilon$ for $i \in [1, N]$, where $\epsilon \sim \mathcal{N}(\mathbf{0}, \mathbf{I})$. However, as depicted in Figure 3, this method still results in temporal inconsistencies between video clips. To tackle this problem, inspired by recent advances in scene composition (Jiménez, 2023), we propose to apply ControlVideo for overlapping short videos and then fuse them together using a defined weight function at each denoising step. This strategy encourages the overlapping frames to merge features from neighboring short videos, effectively mitigating inconsistency issues between adjacent video clips. In the subsequent denoising step, both non-overlapping and overlapping frames within a short video clip are fed into ControlVideo together, which brings the features of non-overlapping frames closer to those of the overlapping frames, thus indirectly improving global temporal consistency. Formally, the $j$-th short video clip $X_t^j$ and the corresponding visual conditions $C^j$ are defined as:

$$X_t^j = \{x_t^i\}_{i=(j-1)(L-a)+1}^{\min((j-1)(L-a)+L,N)}, \quad C^j = \{c^i\}_{i=(j-1)(L-a)+1}^{\min((j-1)(L-a)+L,N)}, \quad j \in [1, n] \tag{6}$$

where $n = \lfloor N/(L-a) \rfloor + 1$ is the number of short video clips, $L$ is the length of short video clip and $a$ is the overlapped length. Let $\epsilon_\theta^j \in \mathbb{R}^{L \times D} = \epsilon_\theta(X_t^j, C^j, p, t)$ denote the ControlVideo for $j$-th short video and $\hat{\epsilon}_\theta \in \mathbb{R}^{N \times D}$ denote the fused ControlVideo for entire video. The fusion function $F(\cdot) : \mathbb{R}^{n \times L \times D} \to \mathbb{R}^{N \times D}$ is defined as follows:

$$\hat{\epsilon}_\theta = F(\epsilon_\theta^1, \ldots, \epsilon_\theta^n) = \text{Sum}(\text{Normalize}(O(w_j \otimes \mathbf{1}_D)) \odot O(\epsilon_\theta^j)), \tag{7}$$

where $w_j \in \mathbb{R}_+^L$ is the weight for the $j$-th short video, $\mathbf{1}_D \in \mathbb{R}^D$ is the vector with all elements being 1, $\otimes$ is Kronecker product, $O(\cdot) : \mathbb{R}^{L \times D} \to \mathbb{R}^{N \times D}$ denote the operation for padding with zero, $\odot$ is the element-wise multiplication, Normalize$(\cdot)$ denote the operation that scales each element in the matrix by dividing it with the sum of all elements in the same position and Sum$(\cdot)$ is the operation for adding the elements at corresponding positions in matrix. In this work, we define normal random variables $w_j \sim \mathcal{N}(L/2, \sigma^2)$. As shown in Figure 3 (row 3), this fusion strategy significantly enhances temporal consistency between short videos.

However, this approach directly fuses nearby videos to ensure local consistency between adjacent video clips, and global consistency for the entire video is improved indirectly during repeated denoising steps. Consequently, as illustrated in Figure 3, temporal consistency deteriorates when video clips are spaced farther apart, exemplified by the degradation of the black car into the green car. In light of these observations, a natural question arises: can we fuse more global features directly to enhance long-range temporal consistency further? To achieve this, we create a keyframe video by incorporating the first frame of each short video segment to provide global guidance directly. ControlVideo is then applied to this keyframe video, which is subsequently fused with the previously obtained $\hat{\epsilon}_\theta$. Formally, let $X_t^K = \{x_t^{(j-1)(L-a)+1}\}_{j=1}^n$ denote the keyframe video and $C^K = \{c^{(j-1)(L-a)+1}\}_{j=1}^n$ denote the corresponding visual conditions. The final model $\epsilon_\theta$ is defined as follows:

$$\epsilon_\theta = wO(\epsilon_\theta^K) + (1 - w)\hat{\epsilon}_\theta, \tag{8}$$

where $w \in [0, 1]$ is the weight, $\epsilon_\theta^K = \epsilon(X_t^K, C^K, p, t)$. The complete algorithm is presented in Algorithm 1. As depicted in Figure 3 (row 4), with the keyframe video fusion strategy, the color of the car is consistently retained throughout the entire video.

## 4 RELATED WORK

**Diffusion Models for Text-driven Image Editing.** Building upon the remarkable advances of T2I diffusion models (Rombach et al., 2022; Ho et al., 2022), numerous methods have shown promising results in text-driven image editing. In particular, several works such as Prompt-to-Prompt (Hertz et al., 2023), Plug-and-Play (Tumanyan et al., 2022) and Pix2pix-Zero (Parmar et al., 2023) explore

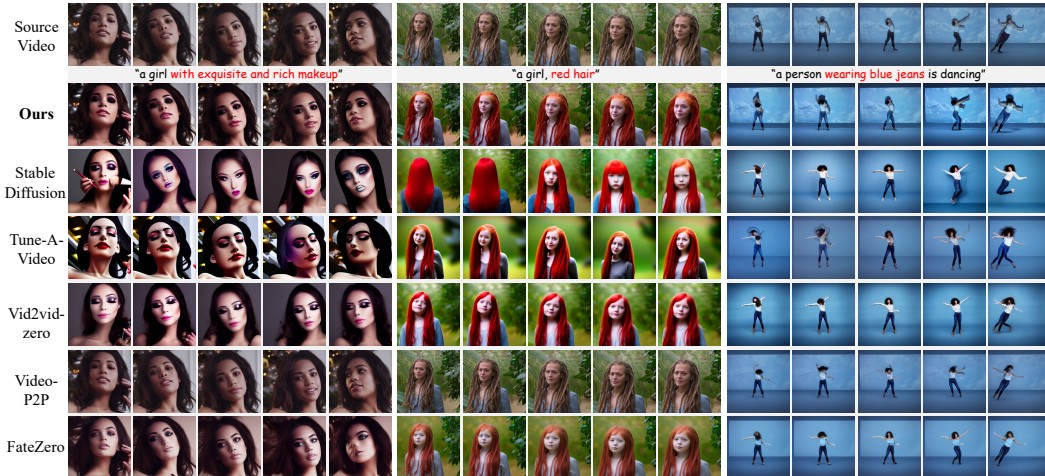

Figure 4: Comparison with baselines. ControlVideo achieves better visual quality by fulfilling three requirements simultaneously. For instance, in the "a girl with red hair" example, ControlVideo not only successfully changes the hair color but also keeps the identity of the female unchanged.

the attention control over the generated content and achieve SOTA results. Such methods usually start from the DDIM inversion and replace attention maps in the generation process with the attention maps from source prompt, which retrain the spatial layout of the source image. Despite significant advances, directly applying these image editing methods to video frames leads to temporal flickering.

**Diffusion Models for Text-driven Video Editing.** Gen-1 (Esser et al., 2023) trains a video diffusion model on large-scale datasets, achieving impressive performance. However, it requires expensive computational resources. To overcome this, recent works build upon T2I diffusion models on a single text-video pair. In particular, Tune-A-Video (Wu et al., 2022) inflates the T2I diffusion model to the T2V diffusion model and finetunes it on the source video-text data. Inspired by this, several works (Qi et al., 2023; Liu et al., 2023a; Wang et al., 2023) combine it with attention map injection methods, achieving superior performance. Despite advances, empirical evidence suggests that they still struggle to faithfully and adequately control the output while preserving temporal consistency.

## 5 EXPERIMENTS

### 5.1 SETUP

For short video editing, following previous research (Wang et al., 2023), we use 8 frames with $512 \times 512$ resolution for fair comparisons. We collect 50 video-text pair data from DAVIS dataset (Pont-Tuset et al., 2017) and website[1]. We compare ControlVideo with Stable Diffusion and the following SOTA text-driven video editing methods: Tune-A-Video (Wu et al., 2022), Vid2vid-zero (Parmar et al., 2023), Video-P2P (Liu et al., 2023a) and FateZero (Qi et al., 2023). For evaluation, following the previous work (Qi et al., 2023), we report CLIP-temp for temporal consistency and CLIP-text for text alignment. We also report SSIM (Wang et al., 2004) within the unedited area between input-output pairs for faithfulness. Additionally, we perform a user study to quantify text alignment, temporal consistency, faithfulness, and overall all aspects by pairwise comparisons between the baselines and ControlVideo. More details are available in the Appendix A.

### 5.2 RESULTS

**Applications.** The main results are shown in Figure 1. Firstly, under the guidance of different *single controls*, ControlVideo delivers videos with high visual realism in attributes, style, and background editing. For instance, HED boundary control helps to change the swan into a Swarovski crystal swan faithfully. Pose control allows shape modification flexibly by changing the man into Sherlock Holmes with a black coat. Secondly, in the "person" → "panda" case, ControlVideo can preserve the

---
[1]https://www.pexels.com

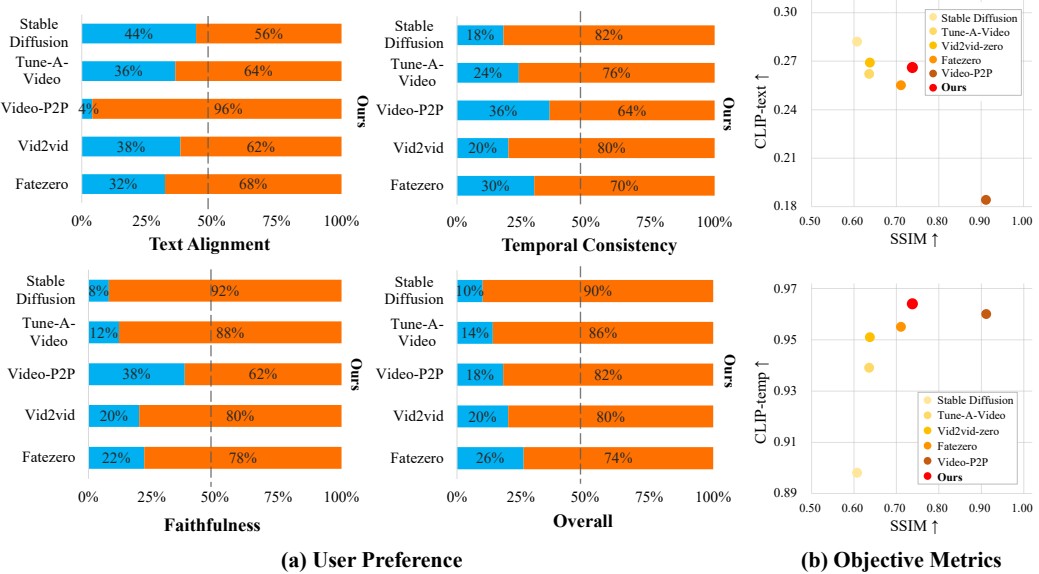

Figure 5: Quantitative results under user study and objective metrics. ControlVideo outperforms all baselines from overall aspects. See detailed analysis in Sec. 5.2.

background and change the shape simultaneously by combining *multiple controls* (Canny edge maps and pose control) to utilize the advantage of different control types. Moreover, in *image-driven video editing*, ControlVideo successfully changes the woman in the source video into Evangeline Lilly to align the reference images. Finally, we can preserve the identity of the woman across hundreds of frames, demonstrating the ability of ControlVideo to maintain *long-range temporal consistency*.

**Comparisons.** The quantitative and qualitative results are shown in Figure 5 and Figure 4 respectively. We emphasize that text-driven video editing should fulfill three requirements simultaneously and a single objective metric cannot reflect the edited results. For instance, Video-P2P with high SSIM tends to reconstruct the source video and fails to align the text. As shown in Figure 4, in the "a girl with red hair" example, it cannot change the hair color. Stable Diffusion and Vid2vid-zero with high CLIP-text generate a girl with striking red hair, but entirely ignore the identity of the female from the source video, leading to unsatisfactory results.

As shown in Figure 5(a), for overall aspects conducted by user study, our method outperforms all baselines significantly. Specifically, 86% persons prefer our edited videos to Tune-A-Video. What's more, human evaluation is the most reasonable quantitative metric for video editing tasks and we can observe ControlVideo outperforms all baselines in all aspects. The qualitative results in Figure 4 are consistent with quantitative results, where ControlVideo not only successfully changes the hair color but also keeps the identity of the female unchanged while all existing methods fail. Overall, extensive results demonstrate that ControlVideo outperforms all baselines by delivering temporal consistent, and faithful videos while still aligning with the text prompt.

**Ablation Studies for Key Components in ControlVideo.** As shown in Figure 3, adding controls provides additional guidance from the source video, thus improving faithfulness a lot. The key-frame attention improves temporal consistency a lot. The temporal attention improves faithfulness and temporal consistency. Combining all the modules achieves the best performance. The quantitative results are shown in Appendix C.

## 6 CONCLUSION

In this paper, we present ControlVideo, a general framework to utilize T2I diffusion models for one-shot video editing, which incorporates additional conditions such as edge maps, the key frame and temporal attention to improve faithfulness and temporal consistency. We demonstrate its effectiveness by outperforming state-of-the-art text-driven video editing methods.

REPRODUCIBILITY STATEMENTS

We submit the code for reproducibility including basic text-driven video editing, image-driven video editing, and long video editing. Please refer to README.md for specific instructions.

ETHICS STATEMENT

We must exercise caution in the application of this method to prevent any potential adverse social consequences, such as the creation of deceptive videos intended to mislead individuals.

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

## A  SETUP

**Implementation Details.** By default, we train the ControlVideo for 80, 300, 500, and 1500 iterations for canny edge maps, HED boundary, depth maps, and pose respectively with a learning rate $3 \times 10^{-5}$. The control scale $\lambda$ is set to 1. For multiple controls, we set $\lambda_i = 0.5$ by default. The DDIM sampler (Song et al., 2020a) with 50 steps and 12 classifier-free guidance are used for inference. The Stable Diffusion 1.5 (Rombach et al., 2022) and ControlNet 1.0 (Zhang & Agrawala, 2023) with canny edge maps, HED boundary, depth maps, and pose are adopted in the experiment. For image-driven video editing, we employ the Lora weight from (Civ) and merge it into Stable Diffusion.

**Evaluation.** The metric for faithfulness only consider the unedited area. The unedited area is computed by Grounding-DINO (Liu et al., 2023b) and SAM (Kirillov et al., 2023) according to text. We evaluate the human preference from text alignment, faithfulness, temporal consistency, and all three aspects combined. A total of 10 subjects participated in this section. Taking faithfulness as an example, given a source video, the participants are instructed to select which edited video is more faithful to the source video in the pairwise comparisons between the baselines and ControlVideo.

**Reproductions.** All baselines are reproduced based on the public code. All used codes in this paper and its license are listed in Table 1.

Table 1: The used codes and license.

| URL | citations | License |
|---|---|---|
| https://github.com/showlab/Tune-A-Video | (Wu et al., 2022) | Apache V2.0 License |
| https://github.com/ChenyangQiQi/FateZero | (Qi et al., 2023) | MIT License |
| https://github.com/baaivision/vid2vid-zero | (Wang et al., 2023) | Unknown |
| https://github.com/ShaoTengLiu/Video-P2P | (Liu et al., 2023a) | Unknown |
| https://github.com/lllyasviel/ControlNet | (Zhang & Agrawala, 2023) | Apache V2.0 License |
| https://github.com/CompVis/latent-diffusion | (Rombach et al., 2022) | MIT License |

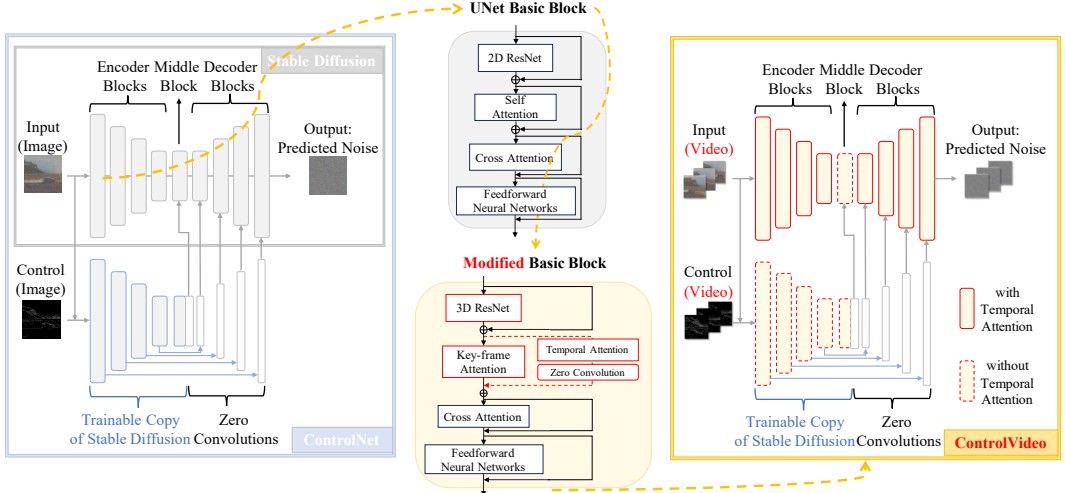

Figure 6: A brief introduction to structures of Stable Diffusion, ControlNet and ControlVideo, highlighted with our modification.

## B    DETAILS OF BACKGROUND

In this section, we describe and visualize (see Figure 6) the detailed architecture of Stable Diffusion and ControlNet. Specifically, the U-Net in Stable Diffusion consists of an encoder, a middle block, and a decoder. The encoder and decoder each consist of 12 blocks, while the full model encompasses a total of 25 blocks. Within these blocks, 8 are utilized for down-sampling or up-sampling convolution layers, and the remaining blocks constitute the basic building blocks. Each basic block is composed of a transformer block and a residual block. The transformer block incorporates a self-attention layer, a cross-attention layer, and a feedforward neural network. The text embeddings, processed by CLIP text encoder, are integrated into the U-Net via the cross-attention layer. As for ControlNet, it introduces an additional trainable copy of the encoder and middle block from Stable Diffusion to incorporate additional conditions. The outputs of ControlNet are then followed by a zero-initialization convolutional layer, which is subsequently added to the features of the U-Net at the corresponding layer.

## C    SYSTEMATIC EMPIRICAL STUDY FOR KEY COMPONENTS OF CONTROLVIDEO

In this section, we conduct a systematical empirical study by analyzing results on 20 video-text pair data and evaluate CLIP-temp, CLIP-text and SSIM. Recognizing that the quantitative results may diverge from human evaluation, we ultimately prioritize human evaluation as our primary measure, while utilizing the quantitative results as supplementary references.

Table 2: Quantitative results about different choices of key and value in self-attention.

| Method | CLIP-text↑ | CLIP-temp↑ | SSIM ↑ |
|---|---|---|---|
| $v^i$ | 0.263 | 0.905 | 0.635 |
| $[v^m; v^i]$ (Qi et al., 2023) | 0.260 | 0.939 | 0.642 |
| $[v^1; v^{i-1}]$ (Wu et al., 2022; Shin et al., 2023) | 0.264 | 0.953 | 0.639 |
| $[v^1; v^{i-1}; v^{i+1}]$ | 0.261 | 0.941 | 0.637 |
| $[v^1; v^i; v^{i-1}; v^{i+1}]$ | 0.261 | 0.955 | 0.648 |
| $v^k, k = 1$ | 0.263 | 0.954 | 0.655 |
| $v^k, k = 3$ | 0.263 | **0.961** | 0.654 |
| $v^k, k = 5$ | 0.261 | 0.958 | **0.657** |
| $v^k, k = 7$ | 0.260 | 0.958 | 0.650 |

Table 3: Quantitative results about fine-tuned parameters of key-frame attention.

| Method | CLIP-text↑ | CLIP-temp↑ | SSIM ↑ |
|---|---|---|---|
| $W^Q$ | 0.253 | 0.951 | 0.634 |
| $W^K, W^V$ | 0.241 | 0.957 | 0.635 |
| $W^Q, W^K, W^V$ | 0.241 | 0.958 | 0.635 |
| $W^Q, W^K, W^V, W^O$ | 0.244 | **0.961** | 0.641 |
| add Lora (Hu et al., 2021) on $W^Q, W^K, W^V, W^O$ | 0.237 | 0.957 | 0.630 |
| $W^O$ | 0.246 | 0.960 | **0.643** |

## C.1 THE DESIGN OF KEY AND VALUE IN SELF-ATTENTION AND FINE-TUNED PARAMETERS

Let $[; ]$ denote the concat operation. We consider using these embeddings as key and value: (1) $v^i$: original spatial self-attention in T2I models. (2) $v^k$, which is our key-frame attention. We select four different key frames. (3) $[v^m; v^i]$ (Qi et al., 2023), where $m = Round(\frac{N}{2})$. (4) $[v^1; v^{i-1}]$ (Wu et al., 2022; Shin et al., 2023). (5) $[v^1; v^{i-1}; v^{i+1}]$, which includes bi-directional information. (6) $[v^1; v^i; v^{i-1}; v^{i+1}]$. As shown in Figure 7, key-frame attention shows the highest temporal consistency, implying that utilizing a key frame to propagate throughout videos is useful. As shown in Table 2, selecting a key-frame as key and value achieves high temporal consistency performance, which is consistent with the qualitative results. There is no significant difference in different key frame selections. In addition, adding the current frame features $v^i$ shows less temporal inconsistency because the $v^i$ contains different information between frames. For example, the color of the car turned red in $[v^m; v^i]$, $[v^1; v^i; v^{i-1}; v^{i+1}]$ following $v^i$ (column 2). Further, we conduct following experiments to investigate finetune which parameters is more useful: (1) $W^Q$. (2) $W^O$. (3) $W^K, W^V$. (4)$W^Q, W^K, W^V$. (5) $W^Q, W^K, W^V, W^O$. (6) add Lora (Hu et al., 2021) on $W^Q, W^K, W^V, W^O$. Based on the results presented in Table 3, we observe that fine-tuning $W^O$ yields superior performance while utilizing fewer parameters, making it our ultimate selection.

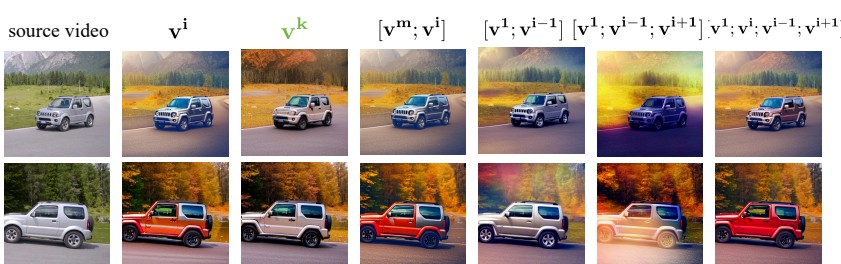

Figure 7: Comparisons with different designs of key and value in self-attention. The green color marked our choice.

Table 4: Quantitative results about different local locations for introducing temporal attention.

| Method | CLIP-text↑ | CLIP-temp↑ | SSIM ↑ |
|---|---|---|---|
| before self-attention | 0.225 | 0.917 | 0.589 |
| after self-attention | 0.241 | 0.909 | **0.629** |
| after cross-attention | 0.241 | 0.902 | 0.616 |
| after FNN | 0.251 | 0.908 | 0.630 |
| with self-attention (random initialization) | 0.230 | 0.909 | 0.585 |
| with self-attention | 0.238 | **0.920** | 0.621 |

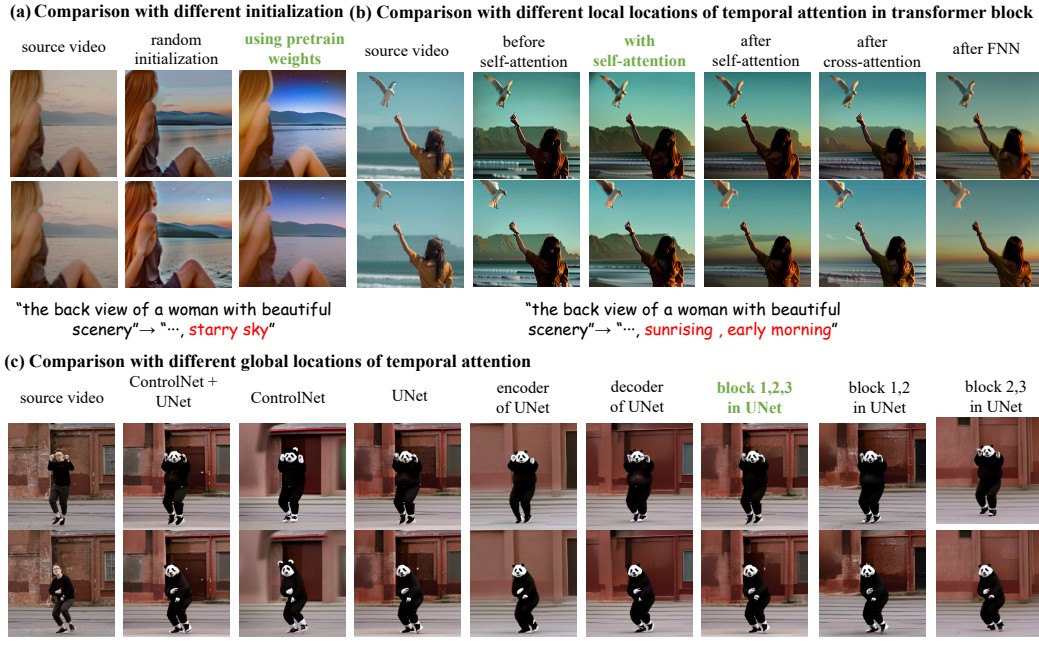

Figure 8: Ablation studies of (a) the way to initialize and the incorporation of (b) local positions and (c) global positions for introducing temporal attention. The green color marked our choice.

## C.2 THE WAY TO INITIALIZATION AND THE INCORPORATION OF LOCAL AND GLOBAL POSITIONS FOR INTRODUCING TEMPORAL ATTENTION

As shown in Figure 8(a), using pretrain spatial self-attention weights as initialization achieves better performance. Next, we explore following potential locations to incorporate temporal attention in transformer blocks: (1) before self-attention. (2) with self-attention. (3) after self-attention. (4)after cross-attention. (5) after FNN. As shown in Figure 8(b), before self-attention and with self-attention result in the best temporal consistency. This is because the input of these two locations is the same as spatial self-attention, which serves as the initial weight of temporal attention. Notably, with self-attention shows higher text alignment, making it our final choice. Moreover, we find the after FNN location yields the worst temporal consistency and should be avoided. The quantitative results are shown in Table 4, which is consistent with the qualitative results.

To investigate the optimal global location for adding temporal attention, we first conduct the following experiments: (1) ControlNet+UNet. (2) ControlNet. (3) UNet. (4) Encoder of UNet. (5) Decoder of UNet. As shown in Figure 8(c), incorporating temporal attention only to the ControlNet fails to preserve the background and removing it does not decrease performance (all vs UNet). This suggests that ControlNet only extracts condition-related features (e.g. pose) and discards the other features (e.g. background), while U-Net, which is used for generation task, preserves all image information. As such, we ultimately choose to add temporal attention to UNet. Additionally, the decoder location

Table 5: Quantitative results about different global locations for introducing temporal attention. These quantitative results diverge from human evaluation and we ultimately prioritize human evaluation as our primary measure and list them as references.

| Method | CLIP-text↑ | CLIP-temp↑ | SSIM ↑ |
|---|---|---|---|
| all | 0.235 | 0.955 | 0.621 |
| controlnet | 0.243 | 0.959 | 0.643 |
| unet | 0.237 | 0.955 | 0.629 |
| encoder | 0.241 | 0.956 | 0.669 |
| decoder | 0.239 | 0.955 | 0.638 |
| Block 1,2 | 0.239 | 0.957 | 0.632 |
| Block 1,3 | 0.237 | 0.954 | 0.640 |
| Block 2,3 | 0.242 | 0.956 | 0.656 |
| Block 1,2,3 | 0.236 | 0.958 | 0.630 |

Table 6: Ablation studies for key components in ControlVideo.

| Method | CLIP-text↑ | CLIP-temp↑ | SSIM ↑ |
|---|---|---|---|
| Stable Diffusion | 0.282 | 0.898 | 0.608 |
| w/o temporal attention | 0.264 | 0.936 | 0.724 |
| w/o key-frame attention | 0.262 | 0.922 | 0.712 |
| w/o control | 0.276 | 0.956 | 0.642 |
| Our full version | 0.266 | 0.964 | 0.738 |

achieves better performance than the encoder. This may be because, in U-Net, the decoder contains more information than the encoder by using skip connections to incorporate features from the encoder. Next, we investigate the location in UNet by following experiments: (1) all; (2) Block 1,2; (3) Block 1,3; (4) Block 2,3; (5) Block 1,2,3, which is UNet except middle block. As shown in Figure 8(c), the Block 1,2,3 shows similar performance with all while with less parameters, which is chosen as the final design. The quantitative results are shown in Table 5, which diverges from human evaluation. We ultimately prioritize human evaluation as our primary measure and list them as references. From human evaluation aspects (see Figure 4 in the main text), we find adding temporal attention on Block 1,2,3 in UNet achieves good performance.

## C.3    ABLATION STUDIES FOR KEY COMPONENTS IN CONTROLVIDEO

As shown in Table 6, the quantitative results are consistent with the qualitative results in the main text. We can observe that introducing additional control mainly contributes to faithfulness a lot. The key-frame attention and temporal attention mainly contribute to temporal consistency and faithfulness.

Table 7: Quantitative results.

| Method | CLIP-text↑ | CLIP-temp↑ | SSIM ↑ |
|---|---|---|---|
| Stable Diffusion(Rombach et al., 2022) | 0.282 | 0.898 | 0.608 |
| Tune-A-Video(Wu et al., 2022) | 0.262 | 0.939 | 0.636 |
| Videp-P2P(Liu et al., 2023a) | 0.184 | 0.960 | 0.912 |
| Vid2vid-zero(Wang et al., 2023) | 0.269 | 0.951 | 0.638 |
| Fatezero(Qi et al., 2023) | 0.255 | 0.955 | 0.711 |
| Ours | 0.266 | 0.964 | 0.738 |

## D COMPARISONS

In this section, we list the automatic metrics in Table 7 as a reference.

