# OpenReview forum: "ControlVideo: Conditional Control for Text-driven Video Editing and Beyond"
_ICLR.cc/2024/Conference — ICLR 2024 Conference Withdrawn Submission_

### Official Review · Reviewer_qeRf · 2023-10-25

**Soundness:** 2 fair
**Presentation:** 2 fair
**Contribution:** 2 fair
**Rating:** 3
**Confidence:** 2

**Summary:**

This papers proposes video editing system referred to as ControlVideo to enhance the fidelity and temporal consistency via incorporating additional conditions.

**Strengths:**

1. Exploration of adding visual conditions on video editing systems

2. Enhanced quality of resulting video for editing.

**Weaknesses:**

As this paper follows the framework of ControlNet that introduces the idea of providing a visual condition (e.g., edge map) to guide editing, merely adding a condition in the video editing system (This idea is already presented in the paper ControlNet) does not give much insight. I am sorry to think that there is no depth to the challenge that this paper is facing.

The current presented studies in this paper seem like applications of ControlNet in the video domain. if the proposed method is the first work that performs an application for video editing, then it must be published and shared in our societies, but this is not the case.

Authors may claim the contribution about proposed attention, but to the best of my knowledge, Prompt-to-Prompt proposes a more general form of attention blending, it seems like following that work. To be specific, The h = h_{u} + \lambda h_{c} seems like one of the attention blending methods, where prompt-to-prompt has already presented these types of blending with a more generalized version. (attention replace, attention refine, attention re-weighting)

Unfortunately, the further proposed long video editing is also not related to the focus of the proposed method. (I mean that is a good issue but not convincing for this paper. I am hoping to meet deeper studies on it as a single paper to tackle that issue.)

What readers may want to read in this paper should be examples as given below:
(1) What was challenging to adding visual condition in video diffusion model and how to solve
(2) What is the proper condition for applying video editing system.
(3) The correlation between visual condition and fidelity.

**Questions:**

My all questions are based on the aforementioned weakness. please give me a rebuttal on that.

---

### Official Review · Reviewer_R3D6 · 2023-10-31

**Soundness:** 3 good
**Presentation:** 3 good
**Contribution:** 2 fair
**Rating:** 5
**Confidence:** 5

**Summary:**

The paper introduces ControlVideo, a tool for text-driven video editing which generates videos that align with given text while preserving the source video's structure. Leveraging a pre-trained text-to-image diffusion model, ControlVideo improves fidelity and temporal consistency by adding conditions like edge maps and fine-tuning key-frame and temporal attention. The method outperforms several baselines, providing high fidelity and temporal consistency in alignment with text. Further enhancements allow ControlVideo to align videos with reference images and extend to long video editing, maintaining consistency across hundreds of frames. Experimental results confirm its capability to produce visually realistic, lengthy videos.

**Strengths:**

1. Generally speaking, the paper is well written, with all ideas being clearly formulated. Quantitative and qualitative analysis have been provided to support the others’ design choices.

2. The application of the proposed framework to the problem of long video editing is appreciated since previous research attempts on this problem is not much. The provided example demonstrates the effectiveness of the proposed pipeline.

**Weaknesses:**

1. I think the author should polish the Related Work section (e.g., “Diffusion Models for Text-driven Video Editing.”) and the experimental section. Although the authors mention Gen-1, one key emphasis of Gen-1 is utilizing additional controls, which is omitted in the section. Moreover, there are other works targeting similar goals like VideoComposer, which shares a lot of similar functionalities and should be compared with in the experiments. The authors are encouraged to enrich this part in order to objectively formulate their contributions.

2. Concerning temporal attention and key-frame attention, the authors’ contributions are rather limited although many empirical studies have been conducted to finalize the best design choice.

3. From my perspective, long video editing is a significant part of this part. However, in the experimental section, the authors did not provide much visualization and in-depth analysis for this. I suggest putting more emphasis on this part.

**Questions:**

Please refer to the weaknesses.

---

### Official Review · Reviewer_ibYi · 2023-10-31

**Soundness:** 3 good
**Presentation:** 3 good
**Contribution:** 2 fair
**Rating:** 5
**Confidence:** 4

**Summary:**

This paper introduces ControlVideo, a text-to-video editing model that utilizes ControlNet for enhanced fidelity and temporal consistency. Incorporating key frame and temporal attention, the model generates video aligned with textual inputs while preserving the original structure. An overlapping fusion strategy is introduced for long video editing. Both the qualitative and quantitative experiments are performed with the baselines.

**Strengths:**

1. The integration of structure guidance through ControlNet is intuitive and interesting.
2. Extensive ablation studies are conducted to explore the effectiveness of each component.
3. The edited videos exhibit great consistency and fidelity to the provided text prompts.

**Weaknesses:**

1. The novelty seems limited. The use of the first frame for key-frame attention has been previously applied in Text2Video-Zero[1], while Make-A-Video[2] has explored temporal attention. The integration of them is a little bit trivial.
2. For long video editing, some existing methods should be compared, such as the hierarchical sampler in NÜWA-XL[3].
3. The results of Video-P2P in Figures 4 and 5 appear wired, especially for text alignment. How about the distribution of the test data, e.g., proportions of object editing, background editing, and style editing?

[1] Khachatryan, Levon, et al. "Text2video-zero: Text-to-image diffusion models are zero-shot video generators." arXiv preprint arXiv:2303.13439 (2023).
[2] Singer, Uriel, et al. "Make-a-video: Text-to-video generation without text-video data." arXiv preprint arXiv:2209.14792 (2022).
[3] Yin, Shengming, et al. "NUWA-XL: Diffusion over Diffusion for eXtremely Long Video Generation." arXiv preprint arXiv:2303.12346 (2023).

**Questions:**

1. The caption "person -> panda" is missing in Fig. 3(b).

---

### Official Review · Reviewer_ZCzg · 2023-11-01

**Soundness:** 2 fair
**Presentation:** 3 good
**Contribution:** 1 poor
**Rating:** 3
**Confidence:** 4

**Summary:**

The paper proposes ControlVideo, a framework for using pre-trained text-to-video latent diffusion for video editing. The proposed method is mostly a mix of existing and well studied work such as ControlNet, LoRA, DINO and Align-your-Latents. The core idea is to start from DDIM noise and then condition the model on additional conditions such as edge maps which should help maintain the "structure" of the original video. Additionally, the authors proposed to use the first frame as the "key-frame" and condition on that as well to help with longer video editings which need overlapping windows to process. Finally, the paper uses temporal attention to improve temporal consistency of the generated video.

**Strengths:**

The paper is well-written and straightforward to comprehend.
The authors extensively drew from previously published work and open-source codebases, while being diligent in attributing credit to the original sources.
The authors conducted a human study to compare the proposed method against others, which is always advantageous in generative video papers, especially considering the lack of reliable qualitative metrics.

**Weaknesses:**

- Limited novelty: The paper heavily relies on previous research for almost every key component of the entire editing pipeline. Building upon previous research is intrinsic to the nature of research, but the "addition" should be both clear and significant. Unfortunately, in this paper's case, I find the contributions to be marginal. The concepts of conditioning on previous signals, addressing inconsistencies in conditioning signals, and employing temporal attentions have all been previously explored, and this paper does not provide any substantial updates in these areas. Instead, it largely borrows these ideas from the domain of image generation and applies them to video, which is not a novel approach. For example, Blattmann et al., in "Align your Latents: High-Resolution Video Synthesis with Latent Diffusion Models," employed a very similar approach (i.e. temporal attention) to enhance temporal consistency on top of Stable Diffusion.

- Missing implementation details: The paper lacks essential implementation details, which could hinder its reproducibility and diminish its impact on future research. Important details, such as the size of additional parameters in LoRA, specifics of temporal attentions, fine-tuning procedures, including optimizers and data, are absent. This omission also raises concerns about the reliability of the comparisons in Figure 5, as the models may differ significantly in terms of size.

- Lack of videos: Surprisingly, for a paper focusing on videos, there is a notable absence of video content. I strongly encourage the authors to present their results, along with comparisons to other models, on an anonymous website for the benefit of both reviewers and readers.

**Questions:**

- Which dataset was used to fine-tune the model?
- The paper proposes "additional visual conditions such as edge maps". But the exact list of controls seems to be missing. Is it only Canny edge maps and key frame and and pose control? did you test more conditions?
- Consider creating an anonymous website with the videos inside. It's hard to judge the quality of improvements without more clear videos.
- Please provide more details on the implementation details.